# Parent Perspectives of Ear Health and the Relationship with Children’s Speech and Language in the Longitudinal Study of Indigenous Children

**DOI:** 10.3390/children10010165

**Published:** 2023-01-14

**Authors:** Anita Morrow, Neil Orr, Kai Nash, Harvey Coates, Cara Cross, John Robert Evans, Hasantha Gunasekera, Samantha Harkus, Linda Harrison, Sharynne McLeod, Catherine McMahon, Katie Neal, Andrea Salins, Rona Macniven

**Affiliations:** 1Department of Linguistics, Faculty of Medicine, Health and Human Sciences, Macquarie University Hearing, Macquarie University, Sydney, NSW 2109, Australia; 2School of Medicine, The University of Western Australia, Perth, WA 6009, Australia; 3Moondani Toombadool Centre, Swinburne University of Technology, Hawthorn, VIC 3122, Australia; 4Children’s Hospital Westmead Clinical School, The University of Sydney, Sydney, NSW 2000, Australia; 5National Acoustic Laboratories, Macquarie University, Sydney, NSW 2109, Australia; 6Macquarie School of Education, Macquarie University, Sydney, NSW 2109, Australia; 7Charles Sturt University, Panorama Ave, Bathurst, NSW 2795, Australia; 8The Shepherd Centre, Newtown, NSW 2042, Australia; 9School of Population Health, Faculty of Medicine & Health, UNSW Sydney, Sydney, NSW 2052, Australia

**Keywords:** Aboriginal and Torres Strait Islander, indigenous, parents, speech, language, communication, hearing, child, preschool, cohort studies

## Abstract

Health and well-being are holistic concepts that are perceived to be inseparable for Aboriginal and Torres Strait Islander peoples. We examined relationships between parent-reported ear symptoms for 787 Indigenous children at two time points (age 2–3 years, age 4–5 years) and two parent-reported speech and language outcomes one year later (age 5–6 years). Most parents (80.2%) reported no concern about their child’s expressive language and (93.8%) receptive language. Binary logistic regression models examined ear health as a predictor of children’s expressive and receptive speech and language adjusting for sociodemographic and health covariates. For children without parent-reported ear symptoms, there were lower odds of parental concern about expressive speech and language (aOR = 0.45; 95% CI 0.21–0.99) and receptive language (aOR = 0.24; 95% CI 0.09–0.62). Parents were less likely to have concerns about the child’s expressive speech and language if their child was female, lived in urban or regional areas, had excellent or very good global health, or had no disability when aged 2–5 years. Since parent-reported ear health and speech and language concerns were related, Aboriginal and Torres Strait Islander children could benefit from culturally safe, strength-based, and family-centered integrated speech, language, and ear health services.

## 1. Introduction

For Aboriginal and Torres Strait Islander peoples, relationships and connection to Country are central to health and well-being. Good continual health comprises spiritual, environmental, cultural, and mental, as well as physical, health and well-being of the whole community [1]. However, there is a divide between the health outcomes of Aboriginal and Torres Strait Islander peoples and those of non-Indigenous Australians due to the ongoing impacts of settler colonialism and structural inequalities [2,3,4] that have impacts across the life-course.

One of the health disparities experienced by Aboriginal and Torres Strait Islander peoples is the high prevalence of middle ear inflammation or infection (otitis media). Otitis media is a common, although preventable, childhood condition [5], subtypes of which are colloquially referred to as ‘glue ear’ or ‘runny ear’ [6]. Otitis media is most common in young children, affecting about 80% of children before the age of three years [6]. However, it is disproportionally more prevalent, long lasting, and acquired earlier and more severe and complicated among Aboriginal and Torres Strait Islander children than non-Indigenous children [6,7,8,9,10]. The World Health Organization considers prevalence rates of chronic suppurative otitis media (CSOM) over 4% to be ‘a serious public health issue’ [11,12]. In Australia, in some communities, CSOM rates, although difficult to quantify, far exceed 4% and have for many years, despite many attempts to address them [11]. Aboriginal and Torres Strait Islander children also experience otitis media earlier, about 76% within the first year of life [7]. Further, otitis media experienced by Aboriginal and Torres Strait Islander children can be difficult to identify because it can be asymptomatic, yet complications may arise due to more severe variations and progressions of the disease [6]. Risk factors for otitis media include low birth weight, malnutrition, household overcrowding, second-hand tobacco exposure, low socioeconomic status, poor hygiene, and remoteness, while early childhood education attendance and breastfeeding are protective factors [2,6,11]. Additionally, Aboriginal and Torres Strait Islander people frequently experience healthcare barriers such as distance and lack of culturally safe services [2,13].

Middle ear disease has long been thought to be associated with poorer speech and language outcomes; however, recent reviews provide conflicting evidence [14]. Yet, a meta-analysis of 32 studies found a small effect size between children with a history of otitis media and those without regarding measures of oral language performance, receptive language, and expressive language [15]. Another meta-analysis of 14 studies examined the impacts of otitis media with effusion (OME) and associated hearing loss on receptive and expressive language in infants and preschoolers and found no or a very small negative association [16]. However, these meta-analyses included studies with non-Indigenous children. Yet, recent findings from an Australian urban cohort study, that included Aboriginal and Torres Strait Islander children, found that children with otitis media at age six years had normal language development scores at age six to ten years, other than a small negative relationship between children with bilateral otitis media at six years and poor rate of receptive vocabulary growth at age 10 years [5,17]. A proposed hypothesis for the relationship between ear health and communication (speech and language) is that the periods of auditory deprivation caused by otitis media during a critical period of cognitive and language development result in delays in speech and language acquisition [18]. Children develop key listening and communication skills in their first three years of life [19,20]. Early neural, cognitive, and language development are dependent on and molded by sensory input [19,21]. Evidence from permanent childhood hearing loss demonstrates that auditory deprivation leads to changes in neural organization of the auditory cortex, impeding speech discrimination and ultimately language acquisition [19,21,22]. As Aboriginal and Torres Strait Islander children commonly experience otitis media earlier and with more severity than non-Indigenous children, this may have a greater impact on speech and language outcomes. Further, as many speak a language other than English as their first language, this may increase the effect of otitis media on speech and language outcomes, especially when measured using English language clinical assessment tools [23,24].

The Footprints in Time: Longitudinal Study of Indigenous Children (LSIC) has been collecting data from up to 1700 Aboriginal and Torres Strait Islander children in Australia in annual waves since 2008 [25,26]. Each annual wave involves interviews in the English language with the children and their parents about many aspects of health, with the intention to investigate what Indigenous ‘children need to have the best start in life, to grow up strong’ and healthy [25,26]. Previously, researchers have examined LSIC data to explore children’s speech and language competence and strategies to support communication [23] and their drawing and vocabulary skills [27]. However, there has been no previous longitudinal examination of ear health and speech and language outcomes within LSIC. Therefore, the present study aimed to investigate: what is the relationship between parent-reported ear symptoms and parent-reported concern regarding (a) their child’s *expressive* and language skills and (b) their child’s *receptive* language skills? It was hypothesized that parents who reported no ear symptoms when their child was between two and six years old would be less likely to report concern about their child’s speech and language skills at age five to seven years after accounting for covariates.

## 2. Materials and Methods

The study used a strengths-based quantitative approach to identify factors associated with positive health outcomes, in keeping with Aboriginal and Torres Strait Islander principles of self-determination that focuses on cultural strengths rather than on deficits [28]. The study authors are a group of Aboriginal and Torres Strait Islander and non-Indigenous researchers with multidisciplinary expertise and experience across Indigenous health, audiology, speech pathology, pediatrics, and education.

### 2.1. Design

The present study is a longitudinal cohort study using waves one to five of LSIC [29]. This Australian Government funded study was conducted by the National Centre for Longitudinal Data (NCLD) under the Department of Services (DSS) and aimed to investigate what Aboriginal and Torres Strait Islander children need to grow up to be healthy and strong [26].

### 2.2. Participants

The participants of the LSIC study were selected using a two-stage purposive sampling design from 11 communities across Australia, varying in levels of remoteness [25]. Aboriginal and Torres Strait Islander children were identified using Australian Government Centrelink and Medicare databases (from the welfare and primary health care system, respectively) and were invited to join the study. Then, snowball sampling methods were used to recruit additional children to increase the sample size. Although the LSIC sample is not necessarily representative of all Aboriginal and Torres Strait Islander Australian children, it is the largest longitudinal study of any indigenous population worldwide and over the years has achieved a 70% retention rate [26].

In 2008, the first wave of data collection occurred with 1671 participants, approximately 10–15% of the total population of Aboriginal and Torres Strait Islander children in the eligible age groups at that time [25]. The study sample includes two cohorts: one younger ‘B cohort’ of children mostly born in 2006–2008 and an older ‘K cohort’ of children mostly born in 2003–2005; at the inception of the study (Wave 1), the B cohort was aged between zero and eighteen months years old, while the K cohort was aged between three and five years.

The present study investigated the ear health of children in LSIC at two time points: when they were approximately two to four years old (B cohort Wave 3; K cohort Wave 1) and then approximately four to six years old (B cohort Wave 4; K cohort Wave 2). Two speech and language outcomes were examined approximately one year later at around five to seven years old (B cohort Wave 5; K cohort Wave 3) [29]. The ages and waves were chosen prior to analysis, to allow the inclusion of the greatest number of participants in the study (Figure 1). Inclusion criteria were children for whom the parent had provided ear symptoms (predictor variable) responses at both time points and responses to both language questions (outcome variable); children with any of these data missing were excluded. While a younger age would have been desirable to investigate the effects of parent-reported ear symptoms in the first year of life on language outcomes, to achieve this would have meant halving the sample size and thus reducing the study power as outcomes from before the age of three years are not available for the K cohort [29]. The sample size of children aged 12 months or less (all from the B cohort) was too small to conduct statistical analyses, so we used both the B and K cohorts in our analysis. We also do not have a measure of the chronicity of runny ears for this age group, as chronic runny ears are indicated by the occurrence of runny ears in the second and third wave of the study at which time the B cohort are aged between two and three years.

### 2.3. Predictor Variable

The present study used parent-reported ear symptoms as the predictive variable. Particularly in the early waves, the key caregiver of the child, labeled ‘Parent 1’ was the main respondent. In the current study, responses from ‘Parent 1’ are referred to as parent reports. Parent 1 was the study child’s mother in over 90% of cases [29]. Parent-reported ear symptoms were identified from the following questions: ‘Has (STUDY CHILD) had any problems with (his/her) ears or hearing in the last 12 months, especially ongoing conditions?’ If the parent responded ‘yes’, they were then asked which of six descriptions of ear problems their child had experienced. These were (1) runny ears, (2) perforated eardrum, (3) total deafness, (4) deaf in one ear, (5) hearing loss/partially deaf, and (6) other ear problems. The variable of interest was ‘runny ears’ as a proxy for parent-reported ear symptoms. The LSIC questionnaire provides prompts for the interviewer to use to clarify the term ‘runny ears’. This includes ‘glue ear’, ‘tropical ear’, ‘chronic suppurative otitis media’, ‘ear infections’, ‘middle ear infection’, ‘fluid in ears’, and ‘may have needed grommets’. We note that several of these prompts relate to otitis media subtypes that are not ‘runny ears’. However, ‘runny ears’ is a colloquial phrase that describes a discharge of pus or fluid draining from a tympanic membrane perforation (e.g., CSOM and acute otitis media with perforation) or acute otitis externa (not common in children). For the purposes of this study, the selection (or non-selection) of ‘runny ears’ reflects Aboriginal and Torres Strait Islander family understanding of ear health rather than representing precise clinical definitions of ear conditions.

As ear symptoms and associated temporary or chronic hearing loss are hypothesized to interfere with critical cognitive, speech, and language development in the first three years of life [19,22], this study used parent reports at age two to six years as a predictor variable (Figure 1). As chronic ear disease and associated long periods of auditory deprivation are hypothesized to negatively impact speech and language development (Al Sagr and Al Sagr 2021), the study linked two consecutive waves of LSIC data (Waves 1 and 2 for K cohort; Waves 3 and 4 for B cohort) to create a variable of accumulative ear symptoms as a proxy of disease chronicity. This included (a) neither years/waves, (b) one year/wave, or (c) both years/waves [6,30].

### 2.4. Outcome Variables

The two outcome variables related to parent-reported expressive and receptive speech and language concern. Speech relates to speech sounds (consonants, vowels, tones, syllables, intonation) and language includes vocabulary (semantics), grammar (Morphophonology), sentence structure (syntax), and discourse. To determine *expressive* speech and language concern, parents were asked, ‘Do you have any concerns (worries) about how (STUDY CHILD) talks and makes speech sounds? (Would you say, yes, no or a little?)’. To determine *receptive* language concern, parents were asked, ‘Do you have any concerns (worries) about how (STUDY CHILD) understands what you say to (him/her)? (Would you say, yes, no, or a little?)’. These questions are from the Parents’ Evaluation of Developmental Status (PEDS) [31] and have been used as outcome variables to examine children’s speech and language status in other Australian population studies [32,33].

The two questions were asked in LSIC waves one to six (except the K cohort was not asked in waves four and six) [29]. The two outcome variables were examined at age five (wave three for K and wave five for B), as at this age, typically developing children are expected to reach most speech and language developmental milestones [34] and to begin primary school. The responses ‘yes’ and ‘a little’ were combined to create a dichotomized variable increasing the strength of the variable for analysis, as exploratory frequency analysis showed that 15.82% and 7.07% respectively, responded with ‘yes’ or ‘a little’, consistent with a previous study [32].

### 2.5. Covariates

As described above, ear symptoms are related to multiple sociodemographic and health factors, including early childhood education attendance, household overcrowding, second-hand tobacco exposure, socioeconomic status, poor hygiene, remoteness, service access, and breastfeeding [6,11]. Risk factors for speech and language developmental delay include being male, low maternal education, low socioeconomic status, family history, low Apgar score, persistent hearing problems, and reactive temperament, whereas, protective factors include persistent and pro-social temperament, and higher maternal well-being [35,36,37]. The current study investigated the availability of these variables in the LSIC dataset to determine whether they could be included as covariates in statistical modeling. Data were sourced from wave three for the B cohort and from wave one for the K cohort, when possible, to account for other known risk factors at the time of the parent-reported ear symptoms, and speech and language outcomes. The covariates investigated in the current study were sex, remoteness (as categorized as three combined categories from the Australian Statistical Geography Classification Remoteness Area 2006 (ASGC) [38,39]), area-level socioeconomic status from the Decile of Relative Indigenous Socioeconomic Outcomes [40] in tertiles [39], global health [41], if the child was ever breastfed, parent smoking status, maternal education, whether parent’s main source of income was from wages/salary or government pension [42], number of children in household, number of people in household, and whether the child had a disability [29]. The disability variables were derived from responses to ‘Has (STUDY CHILD) had any other health problems in the last 12 months, especially ongoing conditions?’ Parents could confirm which of the following answers applied to their child: intellectual disability, specific learning disability, physical disability, neurological disability, psychiatric disability, autism spectrum disorder, acquired brain injury, or other disability. Confirmatory responses from parents were combined. These questions were not asked in the first wave, so data from the second wave were used for both cohorts.

### 2.6. Data Analysis

Data analysis was conducted using IBM SPSS statistical software. First, LSIC waves were combined to create the accumulative ear health predictor variable and to reflect data representative of the same participants at the different time points of focus. This was followed by unadjusted bivariate analysis (Chi-square tests) of the predictor and outcome variables and of the covariates and the outcome variables. A value of *p* < 0.05 was taken as significant. Significant covariates and covariates where an a priori relationship existed were included in the statistical models. Two binary logistic regression models (odds ratio (OR) and a confidence interval of 95%) were used to investigate relationships between parent-reported ear symptoms and parent-reported expressive and receptive language concerns, after adjusting for all other covariates. Using a strengths-based approach [28], the models were set to compare outcomes from the reference category of ‘OM in the two consecutive waves’, to present data of the desired, rather than deficit, outcomes of no reports of concern or worry and good ear health.

### 2.7. Ethics

LSIC has approval from the Australian Institute of Aboriginal and Torres Strait Islander Studies (AIATSIS) Ethics Committee. State and territory and/or regional ethical approval has been obtained for all study sites through state and territory Human Research Ethics Committees or their equivalents. Approval to access the LSIC data was obtained from the Australian Department of Social Services. At the beginning of the study in 2008/2009 parents provided informed consent for themselves and their children into the study.

## 3. Results

Data were available for the ear health predictor variable and the speech and language outcome variables for 787 children (Table 1). A total of 508 (82.6%) parents reported no concern about their child’s expressive speech and language, and 587 (95.4%) parents reported no concern about their child’s receptive language. Over 500 (508; 82.6%) parents reported their children had no ear symptoms in either wave and reported no concern about their child’s expressive speech and language; 107 (17.4%) parents reported their children had no ear symptoms in either wave and reported concern (‘yes’ or ‘a little’) about their child’s expressive speech and language. A total of 587 (95.4%) parents reported their children had no ear symptoms in either wave and reported no concern about their child’s receptive language, and 28 (4.6%) parents reported their children had no ear symptoms in either wave and reported concern (‘yes’ or ‘a little’) about their child’s receptive language.

The results of the unadjusted bivariate analysis (Chi-square tests) are reported in Table 1. These tests demonstrated significant variation between both parental concern for expressive language skills (*p* ≤ 0.005) and parental worry for receptive language skills (*p* ≤ 0.001) by parent-reported ear symptoms in the two waves prior. There was significant variation between sex (*p* ≤ 0.001), remoteness (*p* ≤ 0.001), socioeconomic status (*p* = 0.008), the child’s global health (*p* ≤ 0.001), and parent-reported non-speech disability (*p* = 0.011) for the concern for expressive language skill variable. However, only global health (*p* = 0.022) and parent-reported non-speech disability (*p* = 0.017) were significantly associated with worry for the receptive language skill variable.

Based on the bivariate associations and a priori relationships, sex, remoteness, socioeconomic status, global health, and non-speech disability were included in both the expressive and receptive speech and language binary logistic regression models as covariates. The results of the adjusted models (Table 2) found that for children reported to not have ear symptoms in the two waves, when aged between two and five years, their parents were less likely to report concern about their expressive speech and language skills at between five and six years (OR = 0.45; 95% CI 0.21–0.99). For children reported to not have ear symptoms in the two waves, when aged between two and four years, their parents were less likely to report concern about their receptive language skills a year on when aged between five and six years (OR = 0.24; 95% CI 0.09–0.62). Additionally, parents of male children were twice as likely as parents of females to express concern regarding expressive speech and language skills (OR = 2.34; 95% CI 1.58–3.45). Compared to children living in remote areas, children living in urban (OR = 2.92; 95% CI 1.48–5.76) or regional (OR = 3.96; 95% CI 2.24–6.99) areas were at least twice as likely to not have parents with concerns about their expressive speech and language. Compared to children with fair or poor global health, children with excellent (OR = 0.20; 95% CI 0.07–0.52) or very good (OR = 0.24; 95% CI 0.09–0.64) global health were less likely to have parents who were concerned about their expressive speech and language. In both models, children whose parents reported that they did not have a disability when aged between two and five years, were less likely to have parents reporting concerns about their expressive or receptive speech and language skills at 5–6 years (OR = 0.35; 95% CI 0.13–0.94 and OR = 0.21; 95% CI 0.07–0.67, respectively).

## 4. Discussion

This study aimed to explore the relationship between parent-reported ear symptoms and parent-reported concern regarding their child’s expressive and language skills and their child’s receptive language skills. The results indicate a strong relationship between Aboriginal and Torres Strait Islander children’s parent-reported ear health in the early years of life and parents’ perceptions of their speech and language skills one year later. Parents who reported their child experienced no ear symptoms in two consecutive years (aged between two and five years) had lower odds of reporting concern for their child’s expressive and receptive speech and language (aged between five and six years). The relationship between parent-reported ear health and expressive speech and language was present after adjusting for the covariates of sex, remoteness, socioeconomic status, and the child’s overall (global) health and (non-speech) disability. The relationship between parent-reported ear health and receptive language was present after adjusting for the covariate (non-speech) disability.

Parent-reported ear symptoms were identified as a stronger predictor of receptive language concern than for expressive speech and language concern, although the proportion of parents reporting receptive language concern was lower than for expressive speech and language concern. This relationship could be due to ear symptoms and related hearing loss having a greater impact on receptive language skills (which relies on speech sound audibility) as perceived by parents, while both are impacted by reduced audibility. Additionally, expressive speech and language (e.g., unintelligible speech) could be more noticeable to parents than receptive language issues (i.e., language comprehension).

In the LSIC population of exclusively Aboriginal and Torres Strait Islander children, the prevalence of ear health symptoms and disease was consistent over the first ten years of LSIC, indicating a continuity of ear symptoms [43]. In the Longitudinal Study of Australian Children (a survey of predominantly non-Indigenous children that includes some Aboriginal and Torres Strait Islander children), otitis media experienced in early life was correlated with later hearing problems experienced by both Aboriginal and Torres Strait Islander and non-Indigenous children [44]. The study findings could be explained by ear symptoms being indicative of otitis media leading to fluctuating hearing loss [5,6], which in turn impacts the auditory access and processing in the co-occurring time points. A period of poor sensory input during the early years (between two and five years) can impact cognitive development which then shows impacts on language abilities and acquisition [19,21]. Additionally, the increased prevalence, severity, and longevity of ear disease among Aboriginal and Torres Strait Islander populations and the young age of the sample could be reasons for the relationships we observed [15,16,18,30].

The results of the present study are consistent with the meta-analyses that found a small effect size between children with a history of otitis media regarding measures of oral language performance, receptive language, and expressive language [15,16]. However, Roberts et al. did caution that the findings reflect outcomes for ‘an ”average”, otherwise-healthy child’, and that the impact of OME on language development may be different for ‘at-risk’ children, including those with developmental or language delay, that may disproportionally affect Aboriginal and Torres Strait Islander children (Roberts, Rosenfeld et al., 2004). In addition, these meta-analyses criticized studies that did not control for other determining factors (Casby 2001, Roberts, Rosenfeld et al., 2004). The breadth of topics in the LSIC dataset allowed examination of a variety of relevant covariates to be included in this study. The results of this study do not, however, imply causation as other factors could be contributing to the relationships evident in this study, such as underlying auditory processing abilities independent of hearing loss resulting from ear disease (Sharma, Wigglesworth et al., 2020).

This study also found higher concern for expressive language skills among parents of boys, consistent with a review of PEDS-measured parental concerns [45]. However, the same review found relationships between socioeconomic status and parent concerns that were not evident in the current study but may reflect differences in socioeconomic patterns in Indigenous regions [40,42], as the review examined PEDS across different population groups [45]. We also found a relationship for remoteness and expressive language concern where there was a lower prevalence of concern in remote living families, which may reflect cultural diversity in remote areas and its impact on PEDS relevance [46]. The current study also found that excellent and very good global health predicted concern for expressive language skills among parents, which was also consistent with the review finding of a relationship between global health and PEDS scores [45].

Aboriginal and Torres Strait Islander children have rich cultural and linguistic traditions, and their speech and language abilities are strengthened through family, community, and educational experiences [23]. Cultural and linguistic diversity and cultural safety must be considered in all aspects of speech and language programs and their measurement. A previous study found that parental interpretation of the concept of ‘concern’, using the same PEDS measure as the present study, varies across language and culture [46]. Exploring the relevance of the PEDS for Aboriginal and Torres Strait Islander families is important for future interpretation, although all measures used in LSIC were tested for face validity in Aboriginal and Torres Strait Islander communities, and their selection was guided by experts [26]. Further, to effectively support Aboriginal and Torres Strait Islander children and families, ear health and speech and language services need to be culturally safe, family-centered, and strengths-based [47,48].

### Strengths and Limitations

A strength of the current study is that it is the first to examine the relationship between ear health and speech and language outcomes in LSIC, the largest longitudinal study of Indigenous children worldwide, and although it does not use a random sample, it is a national data collection that covers many aspects of children’s lives [25]. Representative samples are not essential when the research question relates to the relationship between various characteristics and an outcome rather than the precise determination of population prevalence and outcomes. Another strength of the LSIC study is that it was designed and conducted by Aboriginal and Torres Strait Islander researchers with strong community engagement [26]. This analysis took a strengths-based approach, in keeping with Aboriginal and Torres Strait Islander self-determination principles, and focused on cultural strengths and predictors of optimal speech and language outcomes rather than on deficits [28].

It is important when considering parents’ reports of concern regarding their child’s expressive or receptive speech and language to objectively measure the skills of the child due to the phrasing of the question in the study interviews. In addition, using the term ‘runny ears’ may have underestimated the ear health burden, particularly OME, if respondents did not report concerns if the child did not have a perforated tympanic membrane. Chronological ear examinations, hearing assessments, and speech and language assessments at least six months until three years, then annually until at least age six, may result in greater validity but have cost and time burdens that mean they are unsuitable for a large longitudinal study like LSIC. The strength of the agreement between parental concerns about ear symptoms and speech delays might be overestimated given that a parent’s individual characteristics can impact their responses, such as comorbid clinical anxiety or anxious personality traits/temperament. For example, a ‘worried parent’ might observe age-appropriate speech and language behavior but report concern, while another parent’s personality would incline them to not report concern even when they observe delayed expressive and receptive language behaviors. However, previous studies have found that parental reports on the PEDS correlate well with children’s abilities on objective assessments but is best considered to be a screening tool rather than an objective clinical instrument [45]. Future studies could investigate the impact of objectively measured ear symptoms on objective assessments of language development. Along the same line, caution should be taken with the parent-reported ear symptoms predictor variable, which is not specific to levels of severity or frequency due to the phrasing of the question covering a twelve-month period. While ear symptoms can be difficult to determine, parent-reported data gives child ear health data obtained at a lower cost and with less time and burden than clinical measures in this large sample.

## 5. Conclusions

This study with Aboriginal and Torres Strait Islander children and families adds to current evidence supporting a relationship between parent-reported ear symptoms and children’s speech and language outcomes. Importantly, the results from this study show that for children without parent-reported ear symptoms at two time points between the ages of two and five years, there were lower odds of parental concerns regarding their expressive and receptive speech and language skills a year later. Ear symptoms are preventable yet disproportionately affect Aboriginal and Torres Strait Islander children. These findings indicate the continued need to improve the determinants of poor ear health for optimal child health and speech and language development.

## Figures and Tables

**Figure 1 children-10-00165-f001:**
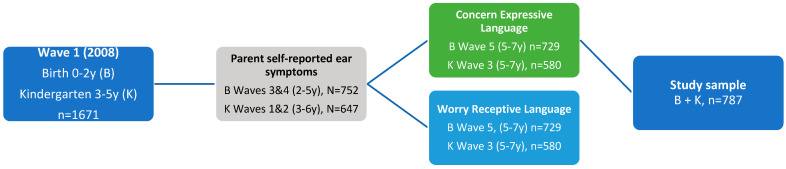
Flow chart of study participants within Footprints in Time: Longitudinal Study of Indigenous Children.

**Table 1 children-10-00165-t001:** Chi-square analysis of covariates by parent-reported concern about expressive and receptive speech and language.

Ear Health and Covariates at Age 2–5 Years by Parent-Reported Concern Regarding Expressive and Receptive Speech and Language Skills at Age 5–6 Years (*N* = 787)						
	Expressive Speech and Language Concern		*p* Value (<0.05)	Receptive Language Concern		*p* Value (<0.05)
	Yes and A Little	No		Yes and A Little	No	*p* = <0.001
**Parent-Reported Ear Symptoms**			***p* ≤ 0.005**		
No waves (0/2)	107 (17.4%)	508 (82.6%)		28 (4.6%)	587 (95.4%)	
One wave (1/2)	36 (27.5%)	95 (72.5%)		14 (10.7%)	117 (89.3%)	
Both waves (2/2)	13 (31.7%)	28 (68.3%)		7 (17.1%)	34 (82.9%)	
**Sex ***			***p* ≤ 0.001**			*p* = 0.330
Male	151 (22.9%)	508 (77.1%)		44 (6.7%)	615 (93.3%)	
Female	91 (14%)	558 (86%)		35 (5.4%)	614 (94.6%)	
**Australian Standard Geographical Classification (ASGC)**			***p* ≤ 0.001**			*p* = 0.66
Urban	60 (21.9%)	214 (78.1%)		15 (5.5%)	259 (94.5%)	
Regional	93 (24%)	295 (76%)		22 (5.7%)	366 (94.3%)	
Remote	32 (10.3%)	279 (89.7%)		22 (7.1%)	289 (92.9%)	
**Indigenous Relative Socioeconomic Outcomes (IRISEO)**			***p* = 0.008**			*p* = 0.626
Most disadvantage	26 (14.4%)	155 (85.6%)		13 (7.2%)	168 (92.8%)	
Mid-advantage	105 (17.9%)	480 (82.1%)		32 (5.5%)	553 (94.5%)	
Most advantage	54 (26.1%)	153 (73.9%)		14 (6.8%)	193 (93.2%)	
**Global Health**			***p* = <0.001**			***p* = 0.022 ^**
Excellent	60 (15.1%)	338 (84.9%)		13 (3.3%)	385 (96.7%)	
Very good	61 (18%)	278 (82%)		28 (8.3%)	311 (91.7%)	
Good	49 (23.9%)	156 (76.1%)		15 (7.3%)	190 (92.7%)	
Fair and poor	14 (48.3%)	15 (51.7%)		3 (10.3%)	26 (89.7%)	
**Ever Breastfed ***			*p* = 0.63			*p* = 0.078
Yes	134 (18.5%)	589 (81.5%)		45 (6.2%)	678 (93.8%)	
No	41 (20.5%)	159 (79.5%)		12 (6%)	188 (94%)	
**Parent Smokes ***			*p* = 0.31			*p* = 0.79
Yes or sometimes	425 (82.2%)	92 (17.8%)		31 (6%)	486 (94%)	
No	327 (79.6%)	84 (20.4%)		27 (6.6%)	384 (93.4%)	
**Maternal Education *****			***p* = 0.026**			*p* = 0.17
High (TAFE cert 3+, uni)	32 (25.8%)	92 (74.2%)		9 (7.3%)	115 (92.7%)	
Medium (Yr11+, TAFE cert I or II)	84 (17.1%)	408 (82.9%)		21 (4.3%)	471 (95.7%)	
**Parents’ Income from Wages/Salary**			*p* = 0.220			*p* = 0.422
No, another source	89 (17.6%)	416 (82.4%)		34 (6.7%)	471 (93.3%)	
Yes, wages or salary	96 (20.8%)	365 (79.2%)		25 (5.4%)	436 (94.6%)	
**Parents’ Income from Government Pension**			*p* = 1			*p* = 0.529
No, another source	44 (19.2%)	185 (80.8%)		16 (7%)	213 (93%)	
Yes, any government pension, benefit, or allowance	141 (19.1%)	596 (80.9%)		43 (5.8%)	694 (94.2%)	
**Number of Children in household**			*p* = 0.144			*p* = 0.178
1–4 children	164 (19.6%)	672 (80.4%)		47 (5.6%)	789 (94.4%)	
5+ children	21 (15.3%)	116 (84.7%)		12 (8.8%)	125 (91.2%)	
**Number of People in Household**			*p* = 0.857			*p* = 0.567
1–4 people	87 (19.8%)	353 (81.5%)		24 (5.5%)	416 (94.5%)	
5–7 people	78 (18.5%)	343 (81.5%)		26 (6.2%)	395 (93.8%)	
8+ people	20 (17.9%)	92 (82.1%)		9 (8%)	103 (92%)	
**Disability ******			***p* = 0.011**			***p* = 0.017 ^**
No	164 (18.5%)	723 (81.5%)		53 (6%)	834 (94%)	
Yes	10 (40%)	15 (60%)		5 (20%)	20 (80%)	

Data for the OM and for the covariates was taken from Wave 3 for the B cohort and from wave 1 for the K cohort, unless stated otherwise. A value of *p* < 0.05 was taken as significant. * Data taken from Wave 1 only for both cohorts. *** Data taken from Wave 3 pertaining only to the B cohort. **** Data from Wave 2 pertaining to both cohorts. ^ Fisher’s exact test for group frequencies less than 5.

**Table 2 children-10-00165-t002:** No concern for expressive or receptive speech and language with predictor OM variable and covariates with adjusted OR and 95% CI. The reference category for each variable is noted in parentheses.

Adjusted OR and 95% CI of Parent-Reported Ear Symptoms with Covariates for No Expressive or Receptive Speech and Language Concern		
	No Expressive Speech and Language Concern OR (95% CI)	No Receptive Language Concern OR (95% CI)
**Parent-Reported Ear Symptoms** (ref both waves; 2/2)		
No WAVES (0/2)	**0.45 (0.21–0.99)**	**0.24 (0.09–0.62)**
One WAVE (1/2)	0.81 (0.35–1.88)	0.57 (0.21–1.57)
**Sex** (ref female)		
Male	**2.34 (1.58–3.45)**	1.15 (0.62–2.11)
**Remoteness (ASGC)** (ref remote)		
Urban	**2.92 (1.48–5.76)**	0.86 (0.32–2.34)
Regional	**3.96 (2.24–6.99)**	0.75 (0.34–1.66)
**Socioeconomic Status (IRISEO)** (ref most advantaged.)		
Most disadvantage	1.00 (0.47–2.16)	1.10 (0.33–3.68)
Mid-advantage	0.70 (0.42–1.18)	0.94 (0.34–1.66)
**Global Health** (ref fair and poor)		
Excellent	**0.20 (0.07–0.52)**	0.45 (0.10–2.04)
Very good	**0.24 (0.09–0.64)**	0.91 (0.22–3.88)
Good	0.38 (0.14–1.03)	0.80 (0.18–3.64)
**Disability** (ref yes)		
No	**0.35 (0.13–0.94)**	**0.21 (0.07–0.67)**

## Data Availability

Data are available to approved researchers from the government, academic institutions, and non-profit organizations. Access LSIC data through the ADA Dataverse (link is external) platform.

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
