# Peer review of "Parent Perspectives of Ear Health and the Relationship with Children’s Speech and Language in the Longitudinal Study of Indigenous Children"

_children, 2023, doi:10.3390/children10010165_

Round 1

Reviewer 1 Report

Overall summary

Data from The Footprints in Time: Longitudinal Study of Indigenous Children considering Aboriginal and Torres Strait Islander Children in Australia were investigated in this study. Parental report of ear symptoms re ‘runny ears’ at 2-5 years was considered as a potential predictor of speech and language concerns (identified through parent report) at 5-6 years.

General Comments:

This is clearly an important issue for the Aboriginal and Torres Strait islander children and families but I feel that the report was not presented as clearly as it could have been for the reader to adequately interpret the study and its findings.

I am not sure that the term “holistic” in the abstract and the introduction in relation to health and wellbeing is easy for the reader – what about something like “perceived to be inseparable”?

I am not familiar with the citation/referencing recommendations for the journal – these would need to be checked.

Overall the nature of the cohort data was difficult to follow. A diagram showing the different waves and ages and how these map to the assessment collection points and research question would be helpful.

I think specifying the research aim more clearly (possibly as a question) with the appropriate objectives that are pursued in the results section would be helpful from the start.

I think a major limitation of this study is the fact that only one of the parent reported concerns about hearing – ‘runny ears’ was chosen for investigation. At this young age, in the absence of objective audiological assessment, it is perceivable that OME could result in any of the other descriptors investigated in the parent report on hearing health. In my view – this could potentially underestimate the potential impact of OME on this population and therefore the results may be misleading. This is discussed to some extent in the limitations but not adequately in my view. There are of course issues about including all the descriptors on hearing health - but then providing readers with the caveat to interpret cautiously would be better than underestimation from my perspective.

Abstract:

See my general points above.

 I think a clear research question with objectives would help present the summary more clearly.

Introduction:

Page 2 –

Line 70 - “Risk factors for otitis media include 70 low birth weight, malnutrition household overcrowding….” Please insert a comma after malnutrition.

Line 72 - And “s, poor hygiene and remoteness while early……..” and comma after remoteness.

Line 78 – it might be helpful to clarify that these reviews are about ear disease in the general population (rather than specifically the population of interest in this paper).

Page 3

Line 101 - “As Aboriginal and Torres Strait Islander children commonly experience otitis media earlier than non-Indigenous children and that many speak a language other than English as their first language, this may impact the effect of otitis media on speech and language outcomes, especially when measured using English language clinical assessment tools (Williams and Jacobs 2009, McLeod, Verdon et al. 2014).”

I think these are two separate points – firstly experiencing otitis media earlier than non-Indigenous children may impact S&L outcomes more significantly than when it occurs later in childhood. Secondly  - assessment that is not in the child’s L1, especially early in development, is not likely to capture an accurate picture of the child’s S&L abilities.

Materials and Methods:

Page 3

Line 125 – please briefly outline what a strengths-based quantitative approach is. I see this on page 11, but it should have been included here.

Line 138 “with varying in” change to “varying in”

Page 4 –

Line 163 – “as outcome from before” – change to “outcomes”

Section 2.3 – paragraph one – I am not sure about the rationale justifying only use of the 1st option ‘runny ears’ – certainly option 2 seems like it would have been appropriate to include – and you could argue for including all options as this is parent report and not audiological findings so some of the other descriptors may actually be used by parents to refer to their child’s hearing issues arising from OME. I hear the argument around family understanding of ear health in this population but it isn’t entirely convincing to me. This is particularly so when considering the arguments presented in the second paragraph in this section.

Line 193 – last sentence in this paragraph – This isn’t easy to follow – a diagram would help.

Section 2.4

Line 199  - did you mean to put ‘morphonology’?

Page 5 –

Line 225 – I would like to see the findings of other more current studies identifying risk and protective factors for speech and language delay considered here as well as the 2010 paper.

Line 2228 – confusing – reference to a diagram as noted previously would help.

Line 245 – not clear how reports of speech disability were used

Page 6 –

Results

Section 3

It would be helpful to revisit the aim/research question at the onset of this section.

Table 1 – It would be clearer to note the name of the predictor variable – ear health – in the title.

Back to this ‘no waves’ ‘one wave’ and ‘both waves’ terminology which is very difficult to follow – does no waves mean data merged from all waves?

This section is very confusing – it seems like there are several objectives that were investigated but these aren’t clearly specified within the design section and then in the results section here. The tables are confusing and difficult to follow. The fact that the various waves etc are not clearly drawn out to the reader makes this even more difficult to follow.

I have no doubt there is some really valuable information in this section but it is relatively impenetrable to me as it is currently presented.

Author Response

Reviewer 1

1. Data from The Footprints in Time: Longitudinal Study of Indigenous Children considering Aboriginal and Torres Strait Islander Children in Australia were investigated in this study. Parental report of ear symptoms re ‘runny ears’ at 2-5 years was considered as a potential predictor of speech and language concerns (identified through parent report) at 5-6 years.

General Comments:

This is clearly an important issue for the Aboriginal and Torres Strait islander children and families but I feel that the report was not presented as clearly as it could have been for the reader to adequately interpret the study and its findings.

Thank you for your helpful review and suggestions.

2. I am not sure that the term “holistic” in the abstract and the introduction in relation to health and wellbeing is easy for the reader – what about something like “perceived to be inseparable”?

We have updated to “Health and wellbeing are holistic concepts that are perceived to be inseparable for Aboriginal and Torres Strait Islander peoples.”

3. I am not familiar with the citation/referencing recommendations for the journal – these would need to be checked.

The paper is formatted for the journal requirements that will be finalised before publication should the paper be accepted.

4. Overall the nature of the cohort data was difficult to follow. A diagram showing the different waves and ages and how these map to the assessment collection points and research question would be helpful.

We have included a flow chart that describes the present study population within the cohort (Figure 1).

5. I think specifying the research aim more clearly (possibly as a question) with the appropriate objectives that are pursued in the results section would be helpful from the start.

We have articulated the research question “what is the relationship between parent-reported ear symptoms and parent-reported concern regarding a) their child’s expressive and language skills and b) their child’s receptive language skills?”

6. I think a major limitation of this study is the fact that only one of the parent reported concerns about hearing – ‘runny ears’ was chosen for investigation. At this young age, in the absence of objective audiological assessment, it is perceivable that OME could result in any of the other descriptors investigated in the parent report on hearing health. In my view – this could potentially underestimate the potential impact of OME on this population and therefore the results may be misleading. This is discussed to some extent in the limitations but not adequately in my view. There are of course issues about including all the descriptors on hearing health - but then providing readers with the caveat to interpret cautiously would be better than underestimation from my perspective.

We have carefully considered the use of the parent reported concerns of ‘runny ears’ from the outset of the study and selected it as the most relevant and accurate as using any, or all, of the six options for “any problems with ears or hearing” as the predictor variable would have been associated with potential bias as the study outcome would have included conditions that were ill-defined or not associated with otitis media. We chose the “runny ears” option as many of the other options would not have been specific to middle ear disease (e.g., the three hearing options), were non-specific (e.g., other ear problems) or too specific (e.g., perforated ear drum). We have updated the limitations to specifically acknowledge that OME may have been underestimated in the use of ‘runny ears’ and have described different measures that may result in greater validity but have cost and time burdens that mean they are unsuitable for a large longitudinal study like LSIC. 

7. Abstract:

See my general points above.

 I think a clear research question with objectives would help present the summary more clearly.

The update to the research question and objectives is now better aligned to the abstract content.

8. Introduction:

Page 2 –

Line 70 - “Risk factors for otitis media include low birth weight, malnutrition household overcrowding….” Please insert a comma after malnutrition.

Line 72 - And “s, poor hygiene and remoteness while early……..” and comma after remoteness.

We have added these commas.

9. Line 78 – it might be helpful to clarify that these reviews are about ear disease in the general population (rather than specifically the population of interest in this paper).

We have clarified that this evidence is from non-Indigenous populations.

10. Page 3

Line 101 - “As Aboriginal and Torres Strait Islander children commonly experience otitis media earlier than non-Indigenous children and that many speak a language other than English as their first language, this may impact the effect of otitis media on speech and language outcomes, especially when measured using English language clinical assessment tools (Williams and Jacobs 2009, McLeod, Verdon et al. 2014).”

I think these are two separate points – firstly experiencing otitis media earlier than non-Indigenous children may impact S&L outcomes more significantly than when it occurs later in childhood. Secondly  - assessment that is not in the child’s L1, especially early in development, is not likely to capture an accurate picture of the child’s S&L abilities.

We have reworded these sentences into two to emphasise these points separately.

As Aboriginal and Torres Strait Islander children commonly experience otitis media earlier than non-Indigenous children, this may have a greater impact on speech and language outcomes. Further, as many speak a language other than English as their first language, this may increase the effect of otitis media on speech and language outcomes, especially when measured  using English language clinical assessment tools.

11. Materials and Methods:

Page 3

Line 125 – please briefly outline what a strengths-based quantitative approach is. I see this on page 11, but it should have been included here.

We have elaborated on the approach “The study used a strengths-based quantitative approach to identify factors associated with positive health outcomes, in keeping with Aboriginal and Torres Strait Islander principles of self-determination that focusses on cultural strengths rather than on deficits”

12. Line 138 “with varying in” change to “varying in”

We have changed to “varying in”

13. Page 4 –

Line 163 – “as outcome from before” – change to “outcomes”

We have changed to “outcomes”

14. Section 2.3 – paragraph one – I am not sure about the rationale justifying only use of the 1st option ‘runny ears’ – certainly option 2 seems like it would have been appropriate to include – and you could argue for including all options as this is parent report and not audiological findings so some of the other descriptors may actually be used by parents to refer to their child’s hearing issues arising from OME. I hear the argument around family understanding of ear health in this population but it isn’t entirely convincing to me. This is particularly so when considering the arguments presented in the second paragraph in this section.

The rationale for this choice is that we are primarily interested in children who have experienced otitis media, and who have hearing loss secondary to that. If we had included all hearing loss options in the analysis, we would be counting an unknown number of children with hearing loss from any cause in the analysis. Further, although 'runny ears' is not representative of all OM subtypes, it is more general and visible than 'perforated ear drum'.

15. Line 193 – last sentence in this paragraph – This isn’t easy to follow – a diagram would help.

We have referred the reader to the Figure 1 diagram for clarity.

16. Section 2.4

Line 199  - did you mean to put ‘morphonology’?

Thank you, we have updated to “Morphophonology”

17. Page 5 –

Line 225 – I would like to see the findings of other more current studies identifying risk and protective factors for speech and language delay considered here as well as the 2010 paper.

We have added two references for a review and meta-analysis (Rudolph 2017) and scoping review (Sansavini et al 2021) to support and supplement these risk and protective factors.

18. Line 228 – confusing – reference to a diagram as noted previously would help.

As mentioned, Figure 1 presents the study sample and details visually.

19. Line 245 – not clear how reports of speech disability were used

We have removed reference to the speech disability variable, it was not used in this study.

20. Section 3

It would be helpful to revisit the aim/research question at the onset of this section.

We consider that this would be better placed at the start of the discussion section rather than results, where we have reiterated the study aims.

21. Table 1 – It would be clearer to note the name of the predictor variable – ear health – in the title.

We have added ear health to the title.

22. Back to this ‘no waves’ ‘one wave’ and ‘both waves’ terminology which is very difficult to follow – does no waves mean data merged from all waves?

No waves means that there were no runny ears reported by parents in either wave 1 or wave 2. The both waves variable (and by extension ‘no waves’) was created by merging data from waves 1 and 2. In the tables we have added a numerical description for clarity - No waves (0/2), One wave (1/2), Both waves (2/2)

23. This section is very confusing – it seems like there are several objectives that were investigated but these aren’t clearly specified within the design section and then in the results section here. The tables are confusing and difficult to follow. The fact that the various waves etc are not clearly drawn out to the reader makes this even more difficult to follow. I have no doubt there is some really valuable information in this section but it is relatively impenetrable to me as it is currently presented.

We trust that the study flow chart and information on study waves and articulation of the research question now frames the design and subsequent results, more clearly.

Reviewer 2 Report

Dear Authors

Congratulations for your work. 

I consider that you need to review the next topics:

1. Spaces between words in lines 83, 84, 104, 113, 131, 145, 360, 411, 413 of the document;

2. Results in lines 274 to 276

3. The number of parents is not coincident with number of child - see results in lines 276 to 284 

4. Topics 2. and 3. constraint Table 1. analyses

5. Try to avoid Table 1. presentation in different pages

Author Response

1. Dear Authors

Congratulations for your work.

I consider that you need to review the next topics:

1. Spaces between words in lines 83, 84, 104, 113, 131, 145, 360, 411, 413 of the document;

 Thank you for your review. We have removed the extra spaces. 

2. Results in lines 274 to 276 The number of parents is not coincident with number of child - see results in lines 276 to 284

We have updated the text to be consistent with the table

3. Topics 2. and 3. constraint Table 1. analyses

Try to avoid Table 1. presentation in different pages

We have placed the start of Table 1 on a new page.

Reviewer 3 Report

Based on the data from Department of Social Services’ The Footprints in Time: Longitudinal Study of Indigenous Children (LSIC), the article examines relationships between parent-reported ear symptoms for Indigenous children at two time points (age 2-3 years, age 4-5 years) and two parent-reported speech and language outcomes one year later (age 5-6 years), showing by binary logistic regression models that children without parent-reported ear symptoms were lower odds of parental concern about expressive speech and language and receptive language. Parents were less likely to be concerned about the child’s expressive speech and language if their child was female; lived in urban or regional areas; had excellent or very good global health; or had no disability when aged 2-5 years. The study based on a large sample of data from interviews with parents and use parental reports as a proxy for occurrence of middle ear inflammation or infection (otitis media) in the early childhood and having difficulties in the area of about expressive speech and language and receptive language.

The article is well written and the study is planned correctly. Hence, the reviewer's comments are mainly editorial and pose questions that arose to him while reading. Some of them are purely editorial, and some are a suggestion for additional analyzes of the data held as a supplement to the argument.

1.The two sentences opening the text - lines 46-51 - look unrelated - the health differences between Aboriginal and Torres Strait Islander peoples and that of non-Indigenous Australians are a fact worth being studied, regardless of their sources (Authors have been emphasizing two sources: impacts of settler colonialism and structural inequalities) therefore does not make sense, since the test reaches at most one of them - access to health care and better living conditions [e.g. actual structural inequalities].

The first sentence, on the other hand, can be read as emphasizing the lesser importance of Aboriginal and Torres Strait Islander peoples than non-Indigenous Australians, which was probably not the intention of the Authors.

2. The authors rightly emphasize that "While a younger age would have been desirable to investigate the effects of parent-reported ear symptoms in the first year of life on language outcomes, to achieve this would have meant halving the sample size" - it is worth checking on this a smaller sample of whether the direction of the results would be congruent.

 3. Sentence line 101-4 [“As Aboriginal and Torres Strait Islander children commonly experience otitis media earlier than non-Indigenous children and that many speak a language other than English as their first language, this may impact the effect of otitis media on speech and language outcomes, especially when measured using English language clinical assessment tools (Williams and Jacobs 2009, McLeod, Verdon et al. 2014).” ] is right and true, but probably not related to the research described by the Authors [and even more so - it does not strengthen their arguments for the need for such research]. The justification for their hypothesis, which goes in a different direction than the results of two later meta-analyzes, is rather the specificity of the sample they study, i.e. access to data on a group with a greater chance of relatively long-term untreated diseases due to poorer access to a doctor and in which it is more common. factor considered by the authors to be the causative factor (ie otitis media). The authors' discussion with previous research is the weakest part of the article - it is worth presenting the samples included in previous meta-analyses in more detail and discussing the possibility that the negative results have sources in the specificity of the data that outweigh the analyzed sample.

4. The description of the division into cohorts and waves - despite the visible efforts of the Authors to make it clear and precise - is difficult to understand. It is worth adding a drawing with a diagram of the cohorts and measurements.

 5. L.127-9. [“The study authors are a group of Aboriginal and Torres Strait Islander and non-Indigenous researchers with multidisciplinary expertise and experience across Indigenous health, audiology, speech pathology, pediatrics, and education”] – please justify the importance of the composition of the investigator group for this kind of study (low in Vesthen factor research – a statistical analysis of secondary data) or remove this remark.

7. 6. Please discuss more details the possible reasons for the differences between your results and previous meta-analysis in the discussion of result section as well.

6.      7. Conclusions need to be divided into two paragraphs: with the description of the data analysis result lines 428-433, and the authors' conclusions as to the significance of these results. The sentence "Speech and language are integral to learning, culture, oral tradition, and storytelling for Aboriginal and Torres Strait Islander children and families" should not be included in the Conclusions (it is NOT a conclusion from the presented study), but rather in the introduction to the description of the research area (Introduction). It is not clear to the reviewer what is the purpose of emphasizing in the conclusions of the study specific traits of appropriate medical care (“as well as the continued need to provide accessibility of timely, culturally safe health support”) - these specific features are not related to the study (i.e. they do not result from the data presented by the authors).

8.    7. 

The last remark is a suggestion only:

9.       Line 410 The strength of the agreement between parental concerns about ear symptoms and speech delays might be overestimated given that a parent’s individual characteristics can impact their responses. This remark is correct and important. Please consult your statistical expert about possibility to statistically check it, as this situation is an analog to common variance problem and should be some statistical tools to check its importance for the study.

May be, for example, taking another disease for analysis or adding some buffer symptom to the analysis as a "control group"? Or doing a parallel analysis only for the declarations of the strong (eliminating "sometimes")?

Author Response

1. Based on the data from Department of Social Services’ The Footprints in Time: Longitudinal Study of Indigenous Children (LSIC), the article examines relationships between parent-reported ear symptoms for Indigenous children at two time points (age 2-3 years, age 4-5 years) and two parent-reported speech and language outcomes one year later (age 5-6 years), showing by binary logistic regression models that children without parent-reported ear symptoms were lower odds of parental concern about expressive speech and language and receptive language. Parents were less likely to be concerned about the child’s expressive speech and language if their child was female; lived in urban or regional areas; had excellent or very good global health; or had no disability when aged 2-5 years. The study based on a large sample of data from interviews with parents and use parental reports as a proxy for occurrence of middle ear inflammation or infection (otitis media) in the early childhood and having difficulties in the area of about expressive speech and language and receptive language.

The article is well written and the study is planned correctly. Hence, the reviewer's comments are mainly editorial and pose questions that arose to him while reading. Some of them are purely editorial, and some are a suggestion for additional analyzes of the data held as a supplement to the argument.

 Thank you for your review.

2. 1.The two sentences opening the text - lines 46-51 - look unrelated - the health differences between Aboriginal and Torres Strait Islander peoples and that of non-Indigenous Australians are a fact worth being studied, regardless of their sources (Authors have been emphasizing two sources: impacts of settler colonialism and structural inequalities) therefore does not make sense, since the test reaches at most one of them - access to health care and better living conditions [e.g. actual structural inequalities].

The first sentence, on the other hand, can be read as emphasizing the lesser importance of Aboriginal and Torres Strait Islander peoples than non-Indigenous Australians, which was probably not the intention of the Authors.

We have edited the first sentence to clarity “For Aboriginal and Torres Strait Islander peoples, relationships and connection to Country are central to health and wellbeing. Good continual health comprises spiritual, environmental, cultural, mental, as well as physical health and wellbeing of whole community.

The description of the divide in health outcomes takes a strength based approach, acknowledging the reasons for health disparities - see the Thurber et al reference.

3. The authors rightly emphasize that "While a younger age would have been desirable to investigate the effects of parent-reported ear symptoms in the first year of life on language outcomes, to achieve this would have meant halving the sample size" - it is worth checking on this a smaller sample of whether the direction of the results would be congruent.

The sample size of children aged 12 months or less (all from the B cohort) was too small to conduct statistical analyses so we used both the B and K cohorts in our analysis. We also do not have a measure of the chronicity of runny ears for this age group as chronic runny ears are indicated by the occurrence of runny ears in the second and third wave of the study at which time the B cohort are aged between 2 and 3 years.  We have added this explanation to the methods.

4. Sentence line 101-4 [“As Aboriginal and Torres Strait Islander children commonly experience otitis media earlier than non-Indigenous children and that many speak a language other than English as their first language, this may impact the effect of otitis media on speech and language outcomes, especially when measured using English language clinical assessment tools (Williams and Jacobs 2009, McLeod, Verdon et al. 2014).” ] is right and true, but probably not related to the research described by the Authors [and even more so - it does not strengthen their arguments for the need for such research]. The justification for their hypothesis, which goes in a different direction than the results of two later meta-analyzes, is rather the specificity of the sample they study, i.e. access to data on a group with a greater chance of relatively long-term untreated diseases due to poorer access to a doctor and in which it is more common. factor considered by the authors to be the causative factor (ie otitis media). The authors' discussion with previous research is the weakest part of the article - it is worth presenting the samples included in previous meta-analyses in more detail and discussing the possibility that the negative results have sources in the specificity of the data that outweigh the analyzed sample.

We have clarified the implications “As Aboriginal and Torres Strait Islander children commonly experience otitis media earlier and with more severity than non-Indigenous children, this may have a greater impact on speech and language outcomes” and emphasised how ‘Further, as many speak a language other than English as their first language, this may increase the effect of otitis media on speech and language outcomes, especially when measured  using English language clinical assessment tools”

In the discussion we have now elaborated on the study samples included in the meta-analyses and how they relate to the present study findings.

5. The description of the division into cohorts and waves - despite the visible efforts of the Authors to make it clear and precise - is difficult to understand. It is worth adding a drawing with a diagram of the cohorts and measurements.

We have included a flow chart that describes the present study population within the cohort (Figure 1).

6. L.127-9. [“The study authors are a group of Aboriginal and Torres Strait Islander and non-Indigenous researchers with multidisciplinary expertise and experience across Indigenous health, audiology, speech pathology, pediatrics, and education”] – please justify the importance of the composition of the investigator group for this kind of study (low in Vesthen factor research – a statistical analysis of secondary data) or remove this remark.

For research involving First Nations people in Australia, the authorship group is very conscious of the importance of having First Nations co-investigators, involved at every stage from design to reporting. This mitigates against the risk that data, even if only secondary analysis of large datasets, may be presented in a way that reflects a particular viewpoint that may not be the viewpoint of First Nations people, and could do more harm than good.

7. Please discuss more details the possible reasons for the differences between your results and previous meta-analysis in the discussion of result section as well.

Our study showed similar results to the meta-analyses, that is we found a small effect of runny ears (OME in the studies) on language, but we have expanded this section to included limitations of one of these meta-analyses.

8. Conclusions need to be divided into two paragraphs: with the description of the data analysis result lines 428-433, and the authors' conclusions as to the significance of these results. The sentence "Speech and language are integral to learning, culture, oral tradition, and storytelling for Aboriginal and Torres Strait Islander children and families" should not be included in the Conclusions (it is NOT a conclusion from the presented study), but rather in the introduction to the description of the research area (Introduction). It is not clear to the reviewer what is the purpose of emphasizing in the conclusions of the study specific traits of appropriate medical care (“as well as the continued need to provide accessibility of timely, culturally safe health support”) - these specific features are not related to the study (i.e. they do not result from the data presented by the authors).

We consider that the conclusions can remain as one paragraph but have removed the mentions of appropriate care and placed "Speech and language are integral to learning, culture, oral tradition, and storytelling for Aboriginal and Torres Strait Islander children and families" to the introduction.

9. The last remark is a suggestion only:

Line 410 The strength of the agreement between parental concerns about ear symptoms and speech delays might be overestimated given that a parent’s individual characteristics can impact their responses. This remark is correct and important. Please consult your statistical expert about possibility to statistically check it, as this situation is an analog to common variance problem and should be some statistical tools to check its importance for the study.

May be, for example, taking another disease for analysis or adding some buffer symptom to the analysis as a "control group"? Or doing a parallel analysis only for the declarations of the strong (eliminating "sometimes")?

Thank you for the suggestion.

We do not believe that it would be possible to control for response bias in this way (our interpretation of your suggestion) as the indicators we would be using to check could also be subject to response bias as they would be from the same informants and data source.

Reviewer 4 Report

This is a useful and interesting study.

Author Response

Thank you

Reviewer 5 Report

In this interesting study Morrow et al, showed that in children without parent-reported ear symptoms at two time points between the 2-5 years, there were lower odds of parental concerns regarding their expressive and receptive speech and language skills after a year in Aboriginal and Torres Strait Islander community. Their strength is the study power, and the weakness is the individual characteristics of the reporting parent’s. This manuscript is well written, the study is well designed, and the data supports the results and conclusion. Authors have used appropriate statistical method to make their conclusion.

However, the material method is not very clear, and authors should consider adding tables to make it clear.

1.      A flow chart of the study design will be helpful to understand the design better.

2.      What is the exclusion and inclusion criteria of the study?

3.      Table with participants cohorts & characteristics can be helpful.

4.      Table or flowchart indicating the predictor & outcome variables will help the reader to interpret the results easily.

5.      Considering this study involves Aboriginal and Torres Strait Islander, authors should mention what languages(s) they have used for questionnaire?

6.      Authors should consider providing a table for the list of covariates.

7.      Line 163 not clear please rephrase.

Author Response

In this interesting study Morrow et al, showed that in children without parent-reported ear symptoms at two time points between the 2-5 years, there were lower odds of parental concerns regarding their expressive and receptive speech and language skills after a year in Aboriginal and Torres Strait Islander community. Their strength is the study power, and the weakness is the individual characteristics of the reporting parent’s. This manuscript is well written, the study is well designed, and the data supports the results and conclusion. Authors have used appropriate statistical method to make their conclusion.

However, the material method is not very clear, and authors should consider adding tables to make it clear.

1. A flow chart of the study design will be helpful to understand the design better.

Thank you for your comments.

The variables are presented in results tables and the authors feel it is unnecessary  to repeat these in the methods, where written descriptions are provided.

We have included a flow chart that describes the present study population within the cohort (Figure 1).

2. What is the exclusion and inclusion criteria of the study?

We have now added this in 2.2 Participants “Inclusion criteria were children for whom the parent had provided ear symptoms (predictor variable) responses at both time points and responses to both language questions (outcome variable); children with any of these data missing were excluded.”

3. Table with participants cohorts & characteristics can be helpful.

Participant characteristics are presented in table 1, add cohort/wave labels

4. Table or flowchart indicating the predictor & outcome variables will help the reader to interpret the results easily.

The flow chart (Figure 1) includes the details and timing of the predictor & outcome variables.  

5. Considering this study involves Aboriginal and Torres Strait Islander, authors should mention what languages(s) they have used for questionnaire?

The questionnaires use English, we have clarified this in the Introduction, paragraph 5.  

6. Authors should consider providing a table for the list of covariates.

The covariates are listed in Table 2.

7. Line 163 not clear please rephrase.

We have rephrased for clarity “While a younger age would have been desirable to investigate the effects of parent-reported ear symptoms in the first year of life on language outcomes, to achieve this would have meant halving the sample size and thus reducing the study power as outcomes from before the age of three years are not available for the K cohort”

Round 2

Reviewer 5 Report

Authors have made a significant effort to address all the points raised in previous revision and have improved the manuscript. 

Minor point: please check the revised manuscript title. i guess there should be "and" between the ear health & the relationship.

Author Response

Authors have made a significant effort to address all the points raised in previous revision and have improved the manuscript. 

Minor point: please check the revised manuscript title. i guess there should be "and" between the ear health & the relationship.

Response: thank you for reviewing the revisions for this paper. We have revised the title to "Parent perspectives of ear health and its relationship with children’s speech and language in the Longitudinal Study of Indigenous Children"
